# Group identification drives brain integration for collective performance

**Enhui Xie[1], Shuyi Zha[1], Yiyang Xu[1], Xianchun Li[1,2,3]\***

[1]Shanghai Key Laboratory of Mental Health and Psychological Crisis Intervention, Affiliated Mental Health Center (ECNU), School of Psychology and Cognitive Science, East China Normal University, Shanghai, China; [2]Institute of Wisdom in China, East China Normal University, Shanghai, China; [3]Shanghai Changning Mental Health Center, Shanghai, China

## eLife Assessment

This timely and **important** study used functional near-infrared spectroscopy hyperscanning to examine the neural correlates of how group identification influences collective behavior. The work provides **solid** evidence to indicate that the synchronization of brain activity between different people underlies collective performance and that changes in brain activity patterns within individuals may, in turn, underlie this between-person synchrony, although the order in which different task stages were completed could not be counter-balanced. This study will be of interest to researchers investigating the neuroscience of social behavior.

**\*For correspondence:**
xcli@psy.ecnu.edu.cn

**Competing interest:** The authors declare that no competing interests exist.

## Abstract

Group identification may influence collective behaviors and result in variations in collective performance. However, the evidence for this hypothesis and the neural mechanisms involved remain elusive. To this end, we conducted a study using both single-brain activation and multi-brain synchronization analyses to investigate how group identification influences collective problem-solving in a murder mystery case. Our results showed that groups with high levels of identification performed better individually compared to those with low identification, as supported by single-brain activation in the dorsolateral prefrontal cortex (DLPFC). Furthermore, high-identification groups also showed enhanced collective performance, supported by within-group neural synchronization in the orbitofrontal cortex (OFC). The DLPFC–OFC connectivity played a crucial role in linking individual and collective performance. Overall, our study provides a two-in-one neural model to explain how group identification affects collective decision-making processes, offering valuable insights into the dynamics of group interactions.

## Introduction

While group decision-making is commonly utilized and can yield positive outcomes, not all groups function effectively, and their performance can vary (*De Wilde et al., 2017*; *Kerr and Tindale, 2003*; *Stasser and Abele, 2020*; *Xie et al., 2023a*). Even when two teams are equally skilled and exchange information, one team often outperforms the other. Interestingly, the winning team typically shows stronger cohesion, while the losing team appears to be more disorganized. Recent studies have highlighted the importance of group identification in determining how verbal information exchange affects collective performance (*Reinero et al., 2021*; *Van Bavel and Cunningham, 2012*; *Xie et al., 2023b*). It appears that differences in group identification may play a role in the varying performance of different groups. Here we ask: How does group identification lead to differences in collective performance? What is the underlying mechanism behind this effect?

Social identity theory suggests that group identification is crucial in shaping collective behavior (*Bicchieri, 2002*; *Gundlach et al., 2006*; *Tajfel and Turner, 1979*). This theory elucidates that group identification refers to a person's attitudes toward a specific group and the emotional meaning associated with belonging to it. Research has consistently shown that stronger group identification is associated with better collective behavior, such as improved decision-making accuracy (*Pärnamets et al., 2020*), effectiveness in team-based interventions (*Reinero et al., 2021*), and overall group cooperation (*Van Bavel and Cunningham, 2012*). This is likely due to the positive impact of increased communication, cooperation, and reduced conflict among group members. Additionally, individuals with a strong sense of group identification are more likely to align their goals with those of the group, leading to increased effort toward achieving objectives (*Dikker et al., 2022*; *Prochazkova et al., 2022*). Based on this, we hypothesize that higher group identification will lead to improved collective performance.

Social cognitive functions, associated with specific brain regions such as the orbitofrontal cortex (OFC) and dorsolateral prefrontal cortex (DLPFC), may play a role in this process. When individuals make decisions within a group, the DLPFC is believed to be involved in coordinating and aligning individual choices with the collective goals and norms (*Goupil et al., 2021*; *Jankovic, 2014*; *Yang et al., 2020*). This brain region is responsible for cognitive regulation and is linked to individual decision-making processes like moral and economic decision-making (*Hu et al., 2015*; *Stallen et al., 2018*). In a group setting, the OFC is important for evaluating the potential outcomes and consequences of different decisions (*Bechara, 2004*; *Izquierdo, 2017*; *Wallis, 2007*). It is thought to regulate group decision-making processes by integrating information about social rewards and values (*Dixon and Christoff, 2014*; *Wallis, 2007*). Based on these findings, we hypothesized that group identification and its impact on collective performance may be tracked by the DLPFC and OFC.

To better comprehend the influence of group identification on collective performance, it is important to uncover the dynamic neural processes involved. Previous research has focused on either single- or multi-brain findings, but social interactions are complex and cannot be fully captured by either approach alone (*Frith and Frith, 2012*; *Jiang et al., 2015*; *Xie et al., 2020*; *Xie et al., 2023a*). In the present study, we combined single- and multi-brain analyses to address this limitation. While recent research has identified both decreased single-brain activation and increased synchronization as drivers of social decision-making (*Cheng et al., 2022*), the interaction between single- and multi-brain activities in driving social decision-making is still not fully understood. *Yang et al., 2020* found that increased brain activation connectivity facilitates the integration of social information into decision-making, while *Xie et al., 2023b* suggested that this connectivity could link two processes (i.e., strategy switching and influence management). Moreover, *Ni et al., 2024* indicated that neural synchronization was linked to the strength of social–emotional communication and connections between individuals. An increase in neural synchronization has also been shown to predict the coordination and cooperation abilities of group members (*Lu et al., 2023*). Therefore, we hypothesize that neural synchronization may be related to group performance. Based on the above, our goal was to explore the regulation of single-brain activation and multi-brain synchronization concerning brain activation connectivity, as well as to incorporate the temporal component to fully represent the dynamic nature of brain processes. By using this innovative approach, we can uncover the intricate interplay between individual and collective neural processes, providing insight into how the brain dynamically responds and adapts during social interactions.

Taken together, we examined the impact of group identification on collective performance in a collaborative problem-solving task. Additionally, we investigated the dynamic neural mechanisms underlying this process.

## Results

### Group identification leads to differences in collective performance

We examined the impact of manipulated group identification on increasing intergroup discrimination. First, a repeated-measures ANOVA on group identification revealed a significant interaction between the levels of group identification and the order of rating group identification ($F_{1, 58}$ = 6.83, p<0.012, $\eta^2_p$ = 0.11; Figure 8C). Post hoc paired *t*-tests revealed that following the manipulation, participants in the High Group Identification condition reported higher group identification (group

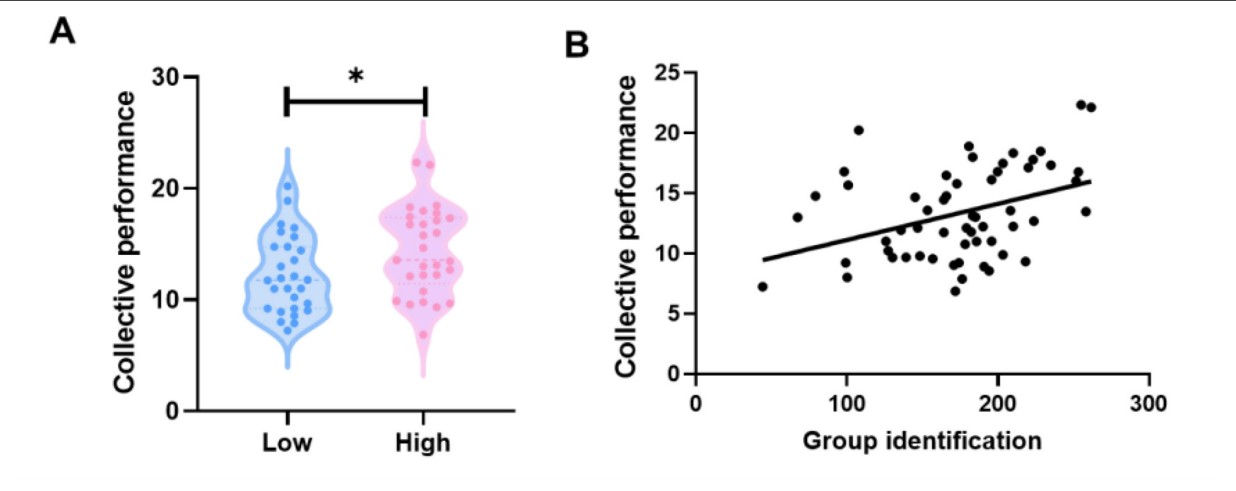

**Figure 1.** Group identification leads to differences in collective performance. (**A**) Manipulated group identification led to group differences in collective performance. (**B**) Group identification was positively correlated with better collective performance. The Pearson's correlation and its associated analyses were based on the data from group identification_2. *p < 0.05.

identification_2) than those in the Low Group Identification condition (group identification_2) ($t_{58}$ = 4.83, p < 0.001). This effect was diminished before manipulation ($t_{58}$ = 1.73, p = 0.090). These results confirmed the effectiveness and validity of our paradigm in inducing different levels of group identification.

Next, we observed significant differences in collective performance between the High and Low groups. Specifically, participants in the High Group Identification group demonstrated a higher level of collective performance ($t_{58}$ = 2.18, p = 0.034, Cohen's $d$ = 0.58) (*Figure 1A*). The results from the regression model highlighted a significant association between the degree of group identification and collective performance ($\beta$ = 0.45, $t$ = 4.56, p = 0.019). Furthermore, the results of Pearson's correlation indicated that groups with higher group identification were more likely to exhibit better collective performance ($r$ = 0.38, p = 0.003) (*Figure 1B*). These findings suggested that manipulated group identification led to variations in collective performance.

## The differences in collective performance are correlated with single-brain activation

We sought to identify single-brain activations that supported individual performance. First, by performing one-sample $t$-tests for single-brain activations, we observed significantly increased single-brain activations in the DLPFC (CH4, $t_{59}$ = 14.54, p < 0.001, false discovery rate [FDR] corrected) and the TPJ (CH1, $t_{59}$ = 3.89, p = 0.022, FDR corrected). Subsequently, we performed independent $t$-tests on single-brain activations in channels that exhibited significant results, with group identification levels as the independent variable. The results showed significant differences in single-brain activations between the High and Low Group Identification groups in the DLPFC (CH4, $t_{58}$ = 2.71, p = 0.015, FDR corrected, Cohen's $d$ = 0.42) (*Figure 2A*).

Furthermore, single-brain activations in the DLPFC (CH4) were associated with individual performance ($r$ = 0.27, p < 0.001, *Figure 2B*), demonstrating that greater single-brain activations in the DLPFC (CH4) were associated with better individual performance. The mediation model demonstrated a satisfactory fit (CFI = 0.93, TLI = 0.93, RMSEA = 0.04) (CFI – Comparative Fit Index; TLI – Tucker–Lewis index; RMSEA – Root-Mean-Square Error of Approximation), suggesting that the perceived group identification of each individual affected the alterations in single-brain activations in the DLPFC, consequently leading to variations in their performance ($\beta_{a}$ = 0.16, $t$ = 2.20, p = 0.030; $\beta_{b}$ = 0.26, $t$ = 3.56, p < 0.001; $\beta_{c}$ = 0.18, $t$ = 2.34, p = 0.020) (*Figure 2C*).

Based on the results mentioned above, the DLPFC (CH4) has been determined as the key neural substrate responsible for individual performance.

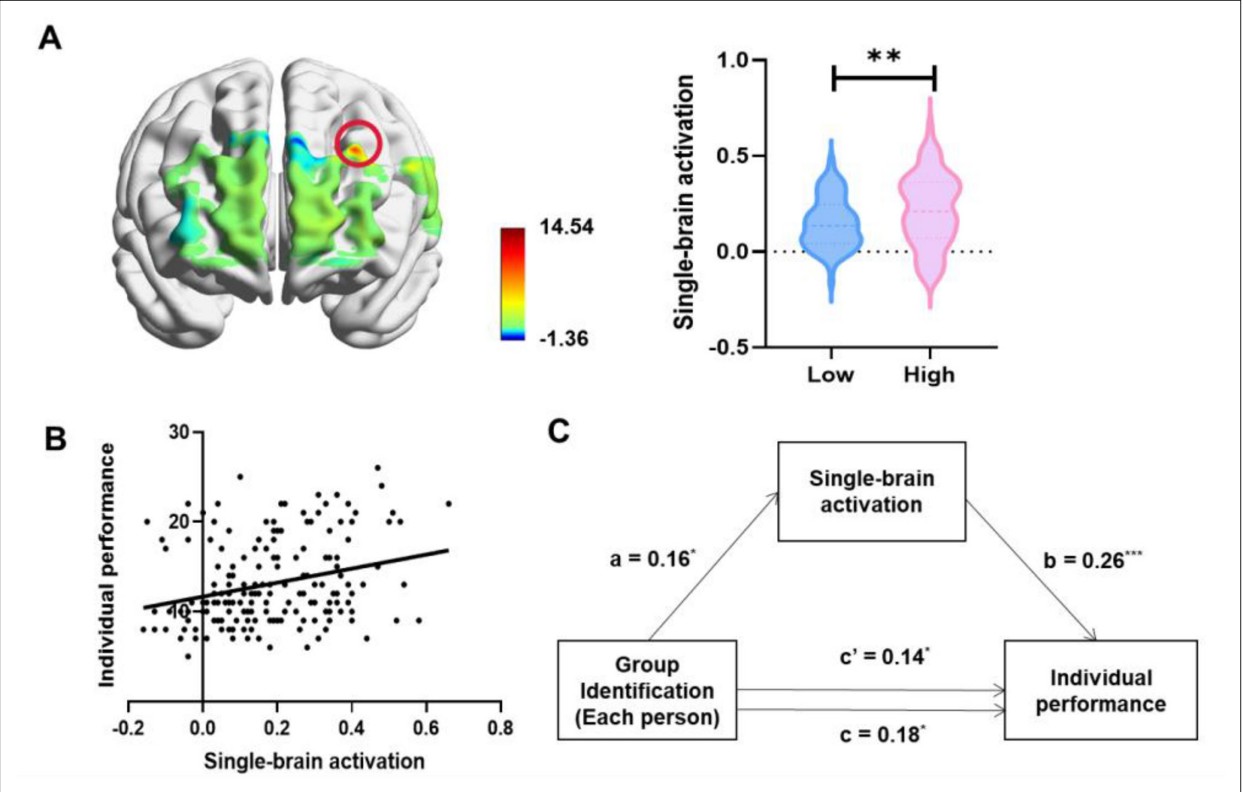

**Figure 2.** The differences in collective performance are correlated with single-brain activation. (**A**) Significant differences in single-brain activations between the High and Low Group Identification groups were observed in the dorsolateral prefrontal cortex (DLPFC) (CH4). (**B**) Greater single-brain activation in the DLPFC (CH4) was associated with higher individual performance. (**C**) A serial mediation model suggested that single-brain activation mediated the relationship between group identification of each person and individual performance. *p < 0.05, **p < 0.01, ***p < 0.001.

## The differences in collective performance are correlated with neural synchronization

We sought to identify neural synchronization that supported collective performance. First, by performing one-sample *t*-tests for group neural synchronization (GNS), we observed a significantly increased GNS in the OFC (CH20, $t_{59}$ = 1.98, p = 0.046, FDR corrected; CH21, $t_{59}$ = 2.49, p = 0.024, FDR corrected) and the TPJ (CH1, $t_{59}$ = 2.16, p = 0.030, FDR corrected). Subsequently, independent *t*-tests were conducted on GNS in the pertinent channels, which revealed that GNS was significantly different between the High and Low Group Identification groups in the OFC (CH21, $t_{58}$ = 2.21, p = 0.037, FDR corrected, Cohen's d = 0.58) (*Figure 3*). A permutation test confirmed that the observed interactive effects on GNS in real groups were outside the 95% CI of a null distribution comprising 1000 pseudo groups (*Figure 3—figure supplement 1*). Therefore, the neural synchronization was only found in the 'real' groups who were interacting in the task. The pattern of associated results was similar to that of HbO when the analyses of HbR were conducted (see the Supplementary Materials).

Results showed that greater GNS in the OFC (CH21) was associated with higher collective performance (*r* = 0.46, p = 0.001; *Figure 3B*). The mediation model demonstrated a satisfactory fit (CFI = 0.86, TLI = 0.86, RMSEA = 0.04), suggesting that group identification of each group caused changes in GNS in the OFC and ultimately impacted the corresponding collective performance ($\beta_a$ = 0.29, SE = 0.01, t = 2.24, p = 0.026; $\beta_b$ = 0.25, SE = 7.28, t = 1.88, p = 0.049; $\beta_c$ = 0.37, SE = 0.01, t = 2.24, p = 0.005) (*Figure 3C*).

Through correlation and regression model analysis, we found that in group decision-making, the increase in group identity would affect group performance by improving GNS in the OFC brain region.

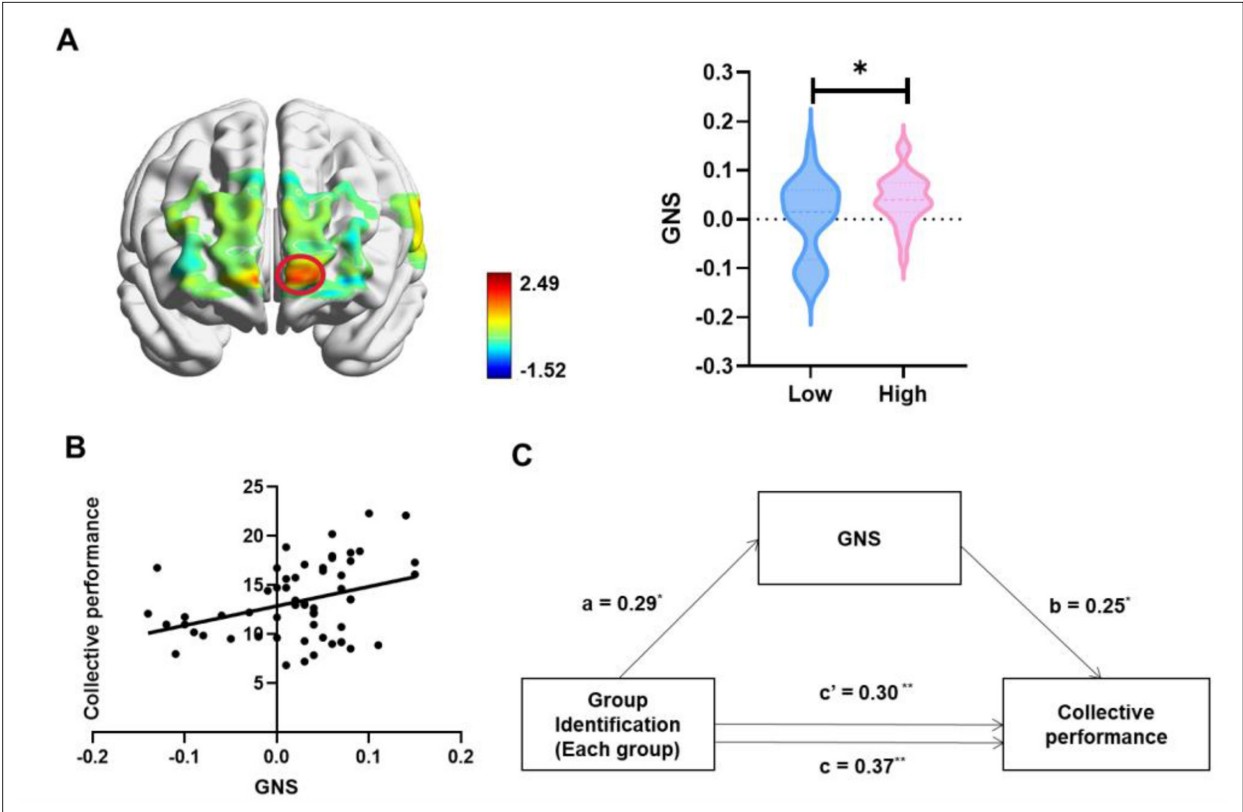

**Figure 3.** The differences in collective performance are correlated with neural synchronization. (**A**) A significant difference between the High and Low Group Identification groups in the orbitofrontal cortex (OFC) (CH21) was observed (p-value, false discovery rate [FDR] corrected). (**B**) Greater group neural synchronization (GNS) in the OFC (CH21) was associated with higher collective performance. (**C**) A mediation model suggested that GNS mediated the relationship between group identification of each group and collective performance. *p < 0.05, **p < 0.01.

The online version of this article includes the following figure supplement(s) for figure 3:

**Figure supplement 1.** A permutation analysis confirmed that the enhanced intergroup coupling shown was group-specific.

## Brain activation connectivity links the single-brain activation and the corresponding GNS

Given the above association between individuals' single-brain activations and individual performance, as well as GNS and collective performance, we subsequently explored how individuals' single-brain activations and the corresponding group's neural synchronizations were linked by brain activation connectivity.

We first extracted the HbO brain activities related to individual performance (e.g., DLPFC, CH4) and collective performance (e.g., OFC, CH21) of each group member and conducted a Pearson's correlation between the two. The outcome of DLPFC–OFC connectivity was then utilized as brain activation connectivity in the following analyses. By performing one-sample $t$-tests, the DLPFC–OFC (CH4–CH21) correlation showed a significant increase during the task ($t_{59}$ = 9.198, p < 0.001, FDR corrected). Subsequently, independent $t$-tests were conducted on DLPFC–OFC connectivity, which revealed that DLPFC–OFC connectivity was significantly different between the High and Low Group Identification groups ($t_{58}$ = 4.01, p < 0.001, FDR corrected, Cohen's $d$ = 0.61) (*Figure 4A*). An independent $t$-test was also conducted on the corresponding behavioral indicators (i.e., the similarity in individual-collective performance) and revealed a significant difference between the High and Low Group Identification groups ($t_{58}$ = 4.03, p < 0.001, Cohen's $d$ = 0.62) (*Figure 4B*). Subsequently, Pearson's correlation was used to test whether individual differences in the similarity in individual-collective performance were reflected by DLPFC–OFC connectivity. It was observed that stronger DLPFC–OFC connectivity was linked to a decrease in the distance between individual and collective performance ($r$ = –0.17, p = 0.026), indicating that stronger DLPFC–OFC connectivity led to a greater similarity in

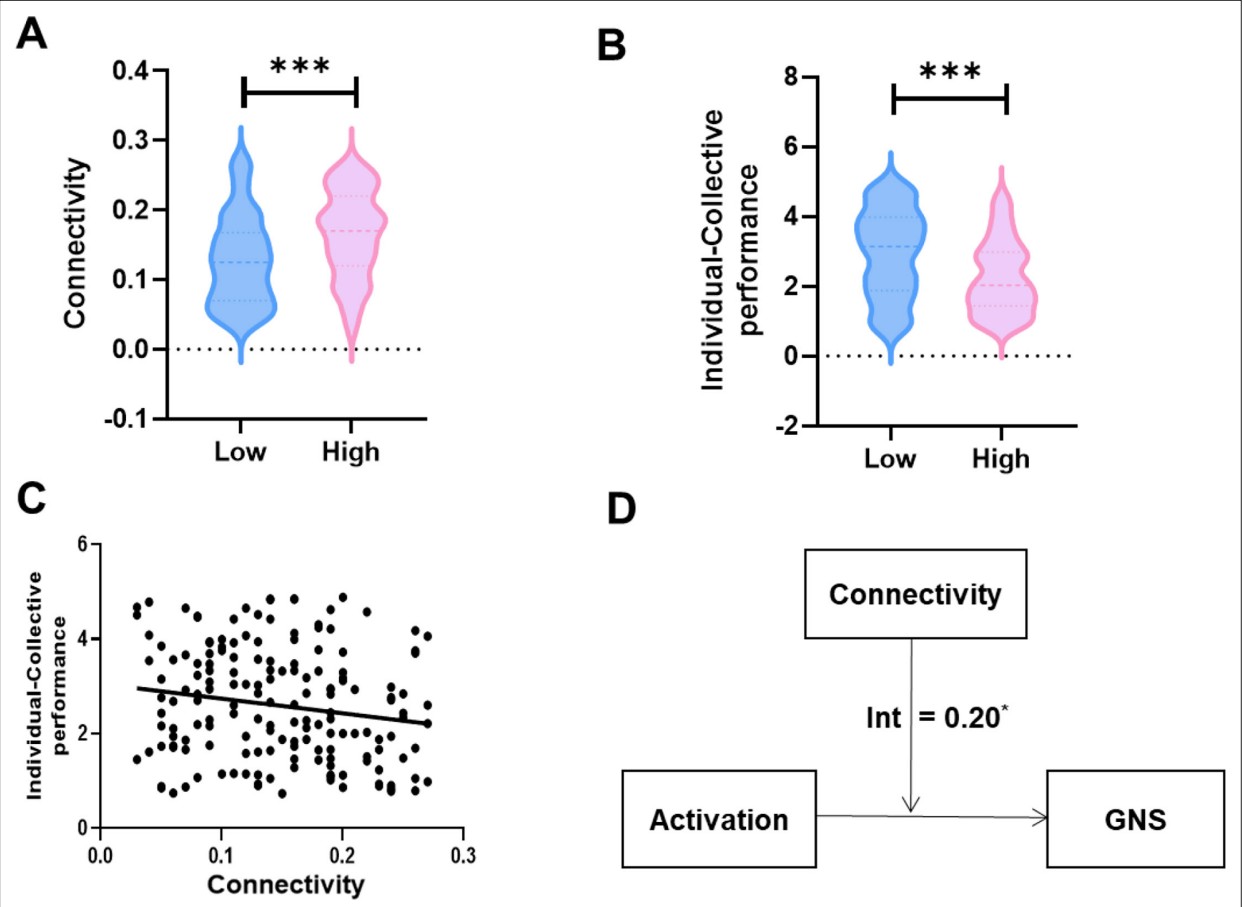

**Figure 4.** Brain activation connectivity links the single-brain activation and the corresponding group neural synchronization (GNS). (**A**) A significant difference between the High and Low Group Identification groups in the dorsolateral prefrontal cortex–orbitofrontal cortex (DLPFC–OFC) (CH4–CH21) correlation was observed (p-value, false discovery rate [FDR] corrected). (**B**) A significant difference between the High and Low Group Identification groups in the similarity in individual-collective performance was observed. (**C**) A stronger DLPFC–OFC (CH4–CH21) correlation was linked to a decrease in the distance between individual and collective performance. (**D**) The analyses suggested that the relationship between individual single-brain activation and the corresponding GNS was regulated by brain activation connectivity. *p < 0.05, ***p < 0.001.

individual-collective performance (*Figure 4C*). Finally, the relationship between individual single-brain activation and the corresponding GNS was regulated by DLPFC–OFC connectivity ($\beta_{int}$ = 0.20, SE = 1.37, $t$ = 2.22, p = 0.028) (*Figure 4D*).

The findings implied that DLPFC–OFC connectivity could link the individual's single-brain activations and the group's GNS, thus unifying individual performance and collective performance.

## The dynamic neural process

To better understand how group identification influences collective performance at a neural level, we examined the dynamic activation of single brains, the GNS, and brain activation connectivity throughout the entire task. Our research findings revealed a significant increase in single-brain activation at approximately 7 min into the task, during the group sharing stage ($t$ = 9.88, p < 0.001; green line in *Figure 5*). Furthermore, there was a notable increase in brain activation connectivity around 12 min into the task, during the group discussion stage ($t$ = 4.70, p = 0.013; black line in *Figure 5*), and GNS at approximately 17 min into the task, also during the group discussion stage ($t$ = 3.01, p = 0.042; orange line in *Figure 5*).

Considering the entire dynamic process, there was a time delay in the transition from individual to collective decisions, and brain activation connectivity was in the middle. In addition to the regulation model provided above, it further supported the processing of information flow in the single-brain activation, DLPFC–OFC connectivity, and ultimately the GNS.

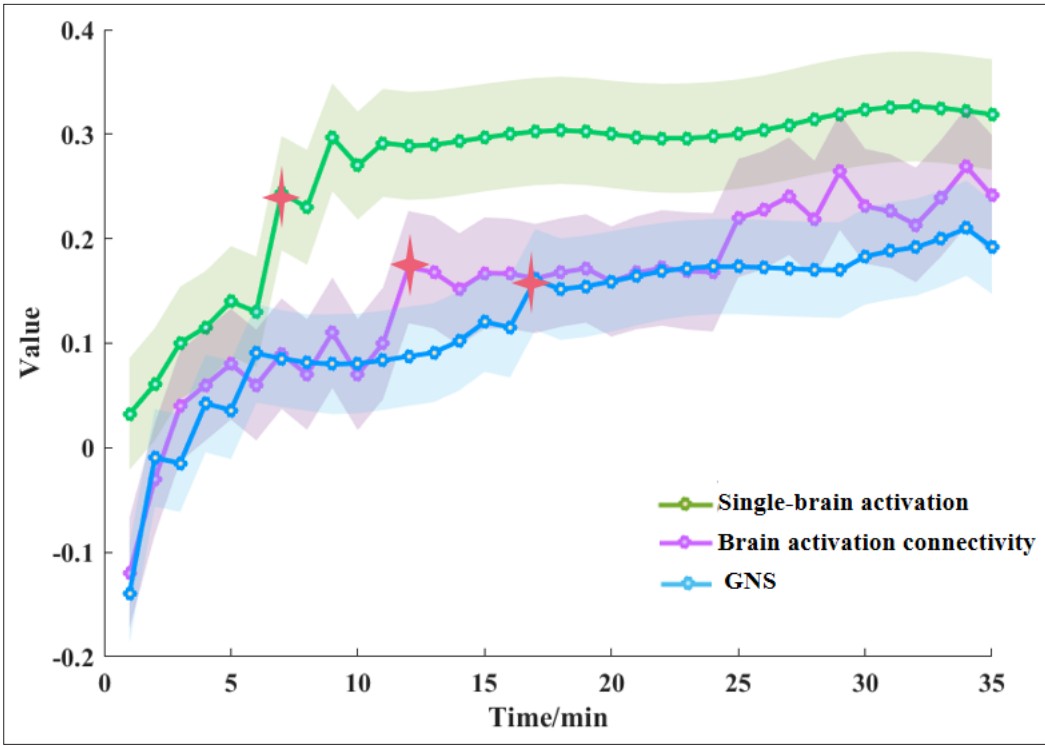

**Figure 5.** The dynamic neural process. There was a time delay in the transition from individual to collective decisions, and brain activation connectivity was in the middle. The processing of information flow in single-brain activation (there was a significant increase in single-brain activation at approximately 7 min into the task), dorsolateral prefrontal cortex–orbitofrontal cortex (DLPFC–OFC) connectivity (there was a significant increase in connectivity at approximately 12 min into the task), and ultimately the group neural synchronization (GNS) (there was a significant increase in GNS at approximately 17 min into the task). The red star sign indicates that at this time point, the neural signal began to increase significantly.

## The two-in-one neural model of group identification influences collective performance

Building on the above results, we have developed a two-in-one neural model that explains how group identification influences collective performance (*Figure 6*). This descriptive model aims to illustrate the potential interrelationships among these indicators and establish a conceptual framework to inspire forthcoming research endeavors. In the first step, group identification influences individual performance, which is associated with significant single-brain activation in the DLPFC of each group member. In the second step, group identification influences collective performance, which is linked to significant within-GNS in the OFC. These two steps are linked by the DLPFC–OFC connectivity, which modulates the relationship between individual DLPFC activation and GNS in the OFC. In particular, people with a strong sense of group identification tend to have better individual performance, which is associated with increased single-brain activation in the DLPFC of each group member. This, in turn, leads to improved collective performance, which is linked to higher levels of within-GNS in the OFC. On the other hand, groups with a low sense of group identification tend to have poorer individual performance, which is associated with decreased activity in the DLPFC. This also results in poorer collective performance, which is correlated with lower levels of GNS in the OFC. The DLPFC–OFC connectivity modulates the relationship between the single-brain activation in the DLPFC and the corresponding group's neural synchronizations in the OFC. This influences the similarity in individual and collective performance, thus connecting individual performance and collective performance.

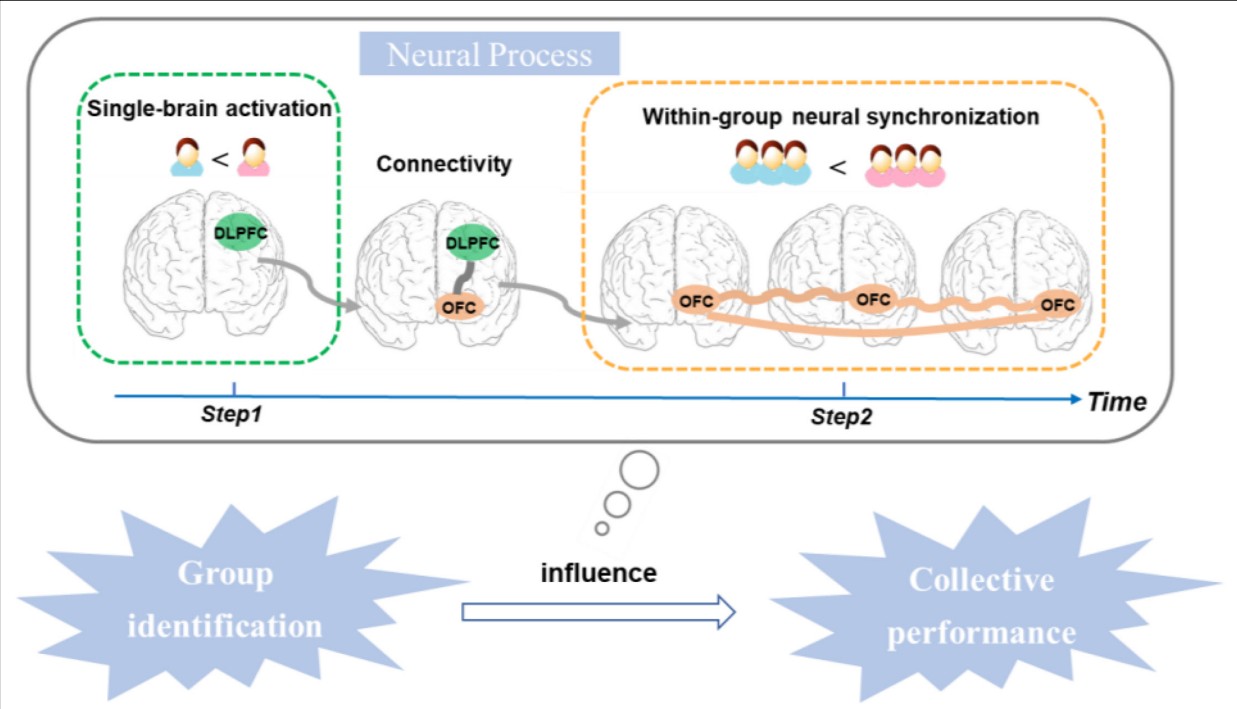

**Figure 6.** The two-in-one neural model explains how group identification influences collective performance. In the first step, group identification influences individual performance, which is associated with significant single-brain activation in the dorsolateral prefrontal cortex (DLPFC) of each group member. In the second step, group identification influences collective performance, which is linked to significant within-group neural synchronization (GNS) in the orbitofrontal cortex (OFC). These two steps are connected by the DLPFC–OFC connectivity, which modulates the relationship between individual DLPFC activation and GNS in the OFC.

### The quality of information exchange is correlated to the effect of group identification

To gain a better understanding of how various group identifications led to differences in collective performance, we gathered further evidence by assessing the quality of information exchange from the video. Independent $t$-tests were conducted on the quality of information exchange, which revealed that the quality of information exchange was significantly different between the High and Low Group Identification groups ($t_{58}$ = 2.53, p = 0.014, Cohen's $d$ = 0.67) (**Figure 7B**). Pearson's correlation showed that the higher quality of information exchange, the better collective performance ($r$ = 0.36, p = 0.007) (**Figure 7C**). The results of our study indicated that interactive frequency played a role in indicating the relationship between group identification and collective performance.

The hierarchical multiple regression analysis showed that the model including both GNS and the quality of information exchange was the most effective in predicting collective performance ($R^2$ = 0.32, SE = 3.33). Additionally, the model with GNS alone ($R^2$ = 0.27, SE = 4.17) outperformed the model with only the quality of information exchange ($R^2$ = 0.24, SE = 4.82) in predicting collective performance. These results indicated that combining GNS and the information exchange quality can accurately predict collective performance. Moreover, GNS proved to be a more reliable predictor of collective performance than behavioral metrics.

### Discussion

Previous research and social identity theory have demonstrated that variations in group identification play a significant role in shaping collective behavior and can impact collective performance (e.g., *Pärnamets et al., 2020*; *Reinero et al., 2021*; *Számadó et al., 2021*; *Solansky, 2011*). Building upon existing findings and theory, our study adds information about the neural mechanisms involved, both at the level of individual group members and at the level of the group. Specifically, we found that group identification influenced individual performance, supported by neural activity in the DLPFC.

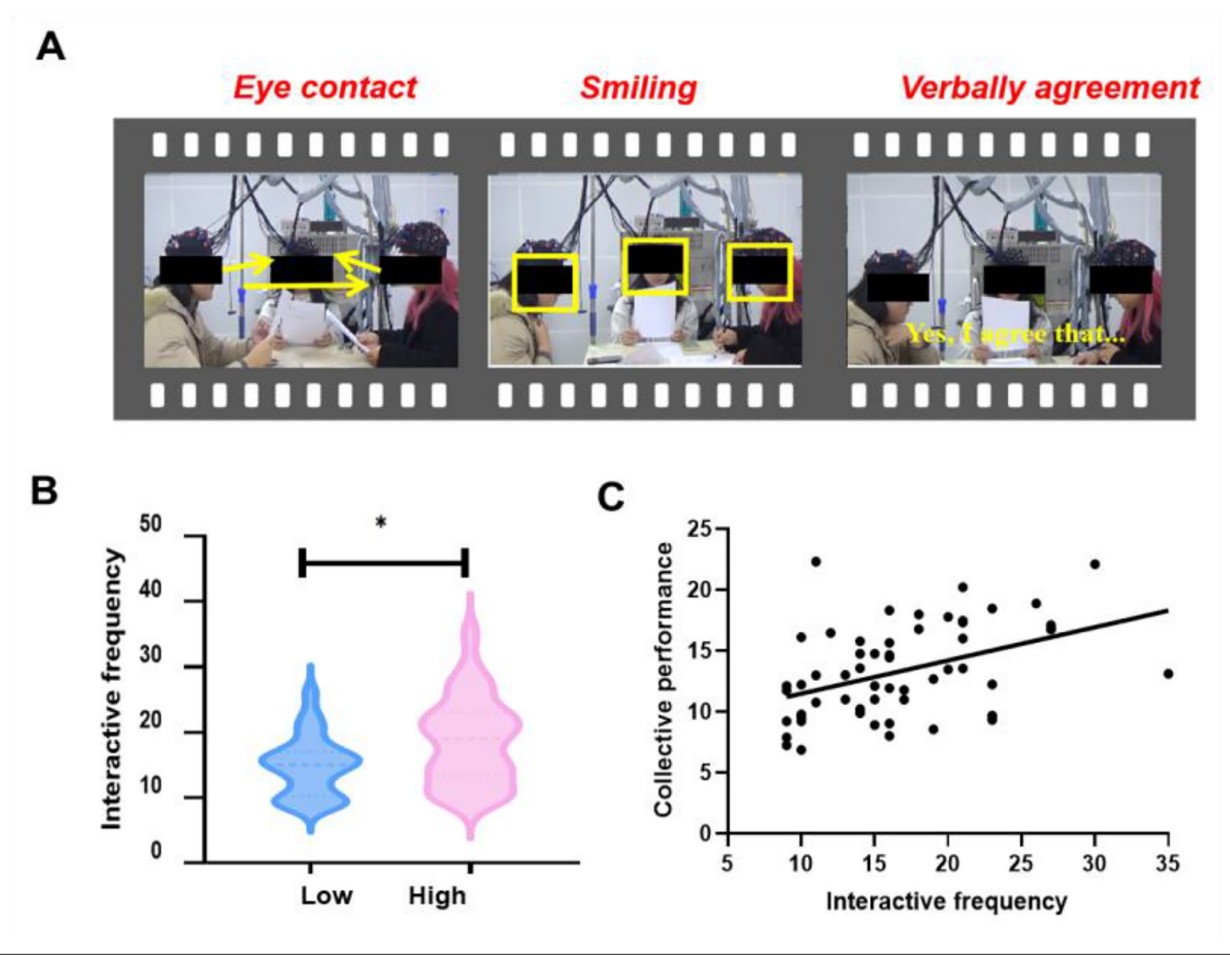

**Figure 7.** The quality of information exchange is correlated to the effect of group identification. (**A**) Assessing the quality of information exchange from the video. (**B**) A significant difference between the High and Low Group Identification groups in the quality of information exchange was observed. (**C**) Higher quality of information exchange was linked to better collective performance. *p < 0.05.

Moreover, we also found that high group identification enhanced collective performance, supported by neural synchronization in the OFC. We highlighted the link between individual performance and collective performance, which was associated with an increased correlation between the DLPFC and the OFC. Based on these results, we have developed a two-in-one neural model to illustrate the dynamic neural processes through which group identification influences collective performance.

Our findings converge with previous work which suggested that group identification is crucial in shaping collective behavior (*Reinero et al., 2021*; *Számadó et al., 2021*; *Solansky, 2011*). Group identification plays a significant role in shaping collective decision-making by promoting cohesion and reducing conflicts within a group (*Bicchieri, 2002*; *Gundlach et al., 2006*; *Tajfel and Turner, 1979*). Members collaborate to address conflicts through positive interactions to uphold positive intergroup relations and achieve better collective behavior (*Brewer and Kramer, 1986*; *De Cremer and Van Vugt, 1999*; *Dikker et al., 2022*; *Prochazkova et al., 2022*). Utilizing tools from neuroscience and video analysis, this study demonstrated the positive impact of increased communication and interaction among group members in groups with high group identification, leading to improved collective performance. These findings support existing research on the influential role of group identification in shaping collective behavior. Understanding how group identification operates can provide valuable insights into group decision-making processes and contribute to the development of strategies for effective group dynamics and decision-making.

Extending previous neuroimaging approaches, this work combined single- and multi-brain approaches to uncover how group identification influenced collective performance: through single-brain activation and multi-brain synchronization. Previous studies have shown that activation in the

DLPFC is associated with controlled decision-making, such as moral and economic decision-making (*Fecteau et al., 2007*; *Gläscher et al., 2012*; *Yang et al., 2020*). Our findings further support this by demonstrating the involvement of DLPFC activation in collective decision-making during social interactions. Additionally, previous research has suggested that increased neural synchronization in the prefrontal area, including the OFC, is linked to collective performance (*Liu et al., 2021*; *Wang et al., 2019*; *Xie et al., 2023a*). Our study expands on this by showing that neural synchrony in the OFC was correlated with collective performance. Interestingly, we found that the level of neural synchrony was higher in the high group identification group compared to the low group, underscoring the significance of group identification in influencing collective performance.

Our research suggests a new perspective on the connectivity between the DLPFC and OFC in the brain, showing its potential role in understanding brain functional integration and the relationship between individual brain activation and collective brain synchronization. While we did not directly determine the reason for this connectivity, it may indicate a form of 'social alignment' between the execution system (individual decision-making) and observation system (collective decision-making) in the human brain (*Adhikari et al., 2013*; *Schenk and Colloca, 2020*; *Yang et al., 2020*). Our study aims to uncover the neural mechanisms behind the impact of group identification on collective performance. We have developed a two-in-one neural model that proposes the connectivity between the DLPFC and OFC as a key mechanism for integrating individual and collective decision-making processes in the brain.

Our study also has demonstrated significant increases in single-brain activation, DLPFC–OFC functional connectivity, and GNS at 7, 12, and 17 min, respectively, following task initiation. The significant increase in these neural activities together constructs the two-in-one neural model that explains how group identification influences the collective performance we proposed. As previous studies have shown, the BOLD signal collected by fNIRS is slowly increasing compared to neuronal activity, which means that it has hysteresis (*Turner et al., 1998*). In social interactions such as group decision-making, the time of neural synchronization is delayed because people need to spend time increasing the number of dialogs to improve collaboration efficiency and form the same preference (*Zhang et al., 2019*). For example, participants would exhibit significant neural alignment, but only after they had completed a period of dialog (*Sievers et al., 2024*). In the task of cooperation, with the improvement of cooperation efficiency between two participants, there is a higher degree of neural synchronization (*Cui et al., 2012*). Therefore, the generation of neural synchronization depends on the interaction over a period of time, which can affect the estimation of collaboration time. Prior research has shown that when the teaching task with prior knowledge began 50–55 s, significant neural synchronization could be generated between teacher and students, which meant that students and teacher achieved the same goal of learning knowledge (*Liu et al., 2019*). Moreover, a noteworthy increase in GNS was observed approximately 6 min into the group discussion period for better discussing and solving the problem (*Xie et al., 2023b*). These findings are similar to ours. Therefore, the time points we found could reflect the dynamic time change of the neural process of team collaboration. Our study has raised several interesting issues for further investigation. First, our laboratory task was specifically created to assess collective performance. While we did observe significant differences in collective performance between groups with high and low levels of identification, future research needs to utilize a trial-by-trial problem-solving task rather than relying on subjective evaluations. This approach would allow for the examination of collective performance as a continuous variable, rather than being reliant on ratings provided by external raters. Second, to enhance the validity of our findings, we utilized fNIRS technology to track both individual and group brain activity simultaneously. Even though our primary focus was on the left temporal regions, specifically lTPJ, we acknowledged that other brain regions such as rTPJ, vmPFC, IFG, and right temporoparietal regions also played a role in Theory of Mind tasks (*Jiang et al., 2015*; *Liu et al., 2023*; *Stolk et al., 2016*) due to the constraints of fNIRS channels. Future studies should consider utilizing MEG technology, which offers better spatial and temporal resolutions, to further investigate the neural mechanisms underlying social cognition and collective behavior. Although our study has found a new perspective, the analysis method still refers to and uses the traditional fNIR-based hyperscanning analyses (*Liu et al., 2019*; *Pärnamets et al., 2020*; *Reinero et al., 2021*; *Számadó et al., 2021*; *Solansky, 2011*), which is generally accepted by the majority of fNIR-based hyperscanning researchers. For example, we would first identify significant channels through a one-sample *t*-test and then conduct

further analyses, such as ANOVA or independent samples *t*-tests. Selective analysis is a powerful tool and is perfectly justified whenever the results are statistically independent of the selection criterion under the null hypothesis (*Kriegeskorte et al., 2009*). However, it may lead to double dipping and missing information. In this study, the absence of statistically significant TPJ activation in the analyzed data led to the TPJ being ignored. In the future, it should be made explicit in the analysis, and the reliability of the results should be ensured by appropriate statistical methods (e.g., cross-validation, independent datasets, or techniques to control for selective bias). Next, our experimental design intentionally separated task phases (reading, sharing, discussing, and decision) to enhance procedural clarity, experimental control, and ecological validity (*Xie et al., 2023a*), departing from the integrated approaches used in prior work (*De Wilde et al., 2017*; *Stasser and Stewart, 1992*). However, the fixed (non-counterbalanced) phase sequence introduces an important limitation: observed effects could potentially reflect temporal factors rather than genuine task-related neural activity. While our analytical approach controlled for some of these temporal confounds, future studies would benefit from implementing counterbalanced phase sequences, jittered trial timing, or additional control tasks to more precisely isolate phase-specific neural correlates. Finally, future research should investigate additional variables, including sex differences and measures of attractiveness or hierarchy among participants, such as students versus teachers.

To summarize, the present study revealed and connected the individual and group brain processes involved in group identification that influenced collective performance and constructed a descriptive model to illustrate this process. This research not only enhances our understanding of the neurobiological aspects of group dynamics and collaboration but also offers a comprehensive framework that synthesizes previous research findings and provides a theoretical foundation for future investigations into the complexities of social interactions.

## Materials and methods
### Participants
G* Power 3.1 (*Faul et al., 2007*) indicates that for *t*-tests with a medium-to-large effect size ($d = 0.70$), an alpha level of 0.05, and a desired statistical power of 0.80 (*Cohen, 1988*), a sample size of at least 52 triads is needed. In total, the data of 60 triads, consisting of 180 healthy college students (96 females, aged 20.45 ± 2.28 years), were included. In addition, 3 participants (2 female, 20.15 ± 1.56 years) without knowledge of the experimental design details were recruited to rate the collective performance, while another 3 participants (2 females, 21.68 ± 1.32 years) were recruited to rate the interactive frequency. The study had full ethical approval by the University Committee on Human Research Protection (HR2-0189-2022), East China Normal University. Informed written consent was obtained from each participant before each experiment.

### Experimental procedures and tasks
The study aimed to investigate how collective performance was affected when group identification was either high or low. Participants came to the laboratory and were randomly assigned to 60 groups, each consisting of 3 people of the same gender. Each triad consisted of participants who were unfamiliar with one another and had not previously engaged in similar tasks. They were randomly assigned to either the high or low group identification manipulation and completed the task while undergoing hyperscanning with fNIRS.

#### Group identification manipulation
To manipulate group identification (High Group Identification and Low Group Identification conditions), we referred to previous studies (*Efferson et al., 2008*; *Yang et al., 2020*). For each High Group Identification condition, the participants in each triad were invited to chat with each other to introduce themselves and find three-person-shared features for minutes. For each Low Group Identification condition, the participants in each triad were asked to chat with each other about the main courses they had been taking during this semester without being explicitly asked to find shared features (*Figure 8A*). The participants rated the group identification in their group before and after the manipulation (i.e., group identification _1 and group identification _2) (*Figure 8B*). Both discussions were conducted for the same duration of 3 min, ensuring that the number of exchanges between the two

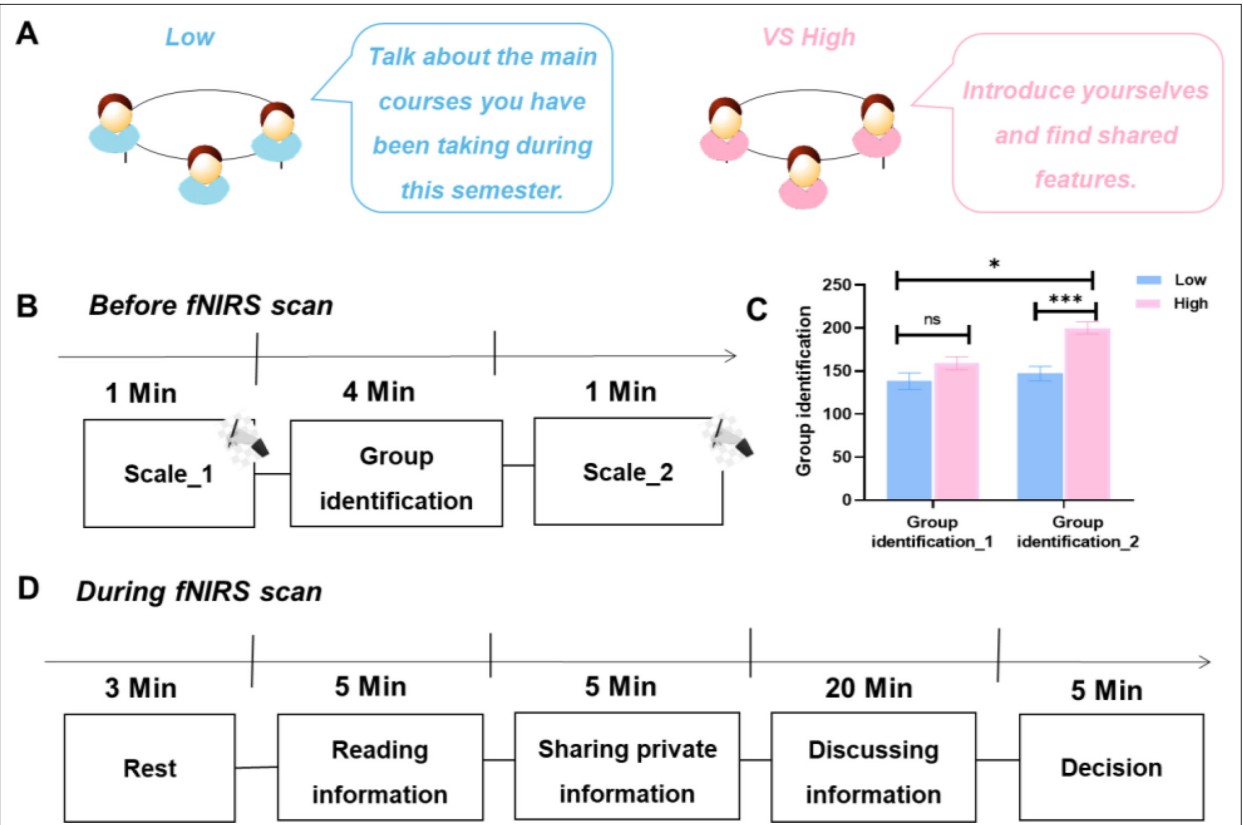

**Figure 8.** Experimental procedure. (**A**) Group identification manipulation. For each High Group Identification condition, the participants in each triad were invited to chat with each other to introduce themselves and find three-person-shared features for minutes. For each Low Group Identification condition, the participants in each triad were asked to chat with each other about the main courses they had been taking during this semester without being explicitly asked to find shared features. (**B**) The rate of group identification. The participants in each triad rated the group identification in their group before and after the manipulation. (**C**) Group identification manipulation check. We examined how the level of group identification changed when we manipulated it, for both High and Low Group Identification conditions. Group identification_1, Group identification before manipulation; Group identification _2, Group identification after manipulation. (**D**) The procedure of task. First, participants completed a series of individual difference questionnaires before the task. Then, each group member received 18 common information and 2 private information. They read their information within 5 min. After that, each triad was required to complete verbal information exchange, comprising both group sharing and group discussion. Each group member texted the private information to other members by Tencent Meeting during group sharing for 5 min, and they discussed the information currently disclosed with others orally during the group discussion for 20 min. Ultimately, the groups had a period of 5 min to answer the questions. *p < 0.05, ***p < 0.001.

The online version of this article includes the following figure supplement(s) for figure 8:

**Figure supplement 1.** Probe location and measure the brain activity simultaneously.

groups remained comparable. The group identification was assessed using a 3-item questionnaire with each item being rated on a 100-point scale ranging from 0 = strongly disagree to 100 = strongly agree (***Van Bavel and Cunningham, 2012***). The questionnaire was presented on participants' phones. The reliability of this scale was confirmed to be high ($\alpha$ = 0.87). The three participants conducted face-to-face offline interaction throughout the manipulation process. In addition to explaining the next phase of the task and controlling the timer, experimenters would be isolated from participants.

## Task

We adopted a well-validated collaborative problem-solving task that incorporated the hidden profile task to solve a murder mystery case (***De Wilde et al., 2017***; ***Stasser and Stewart, 1992***; ***Xie et al., 2023b***). The case description contained 24 relevant arguments that were either incriminating or exonerating for each suspect (Suspects A, B, and C). In total, each suspect had six incriminating arguments presented against them. Additionally, Suspects B and C each had three exonerating arguments to their defense, while Suspect A did not. Accordingly, when combining all 24 relevant

arguments, Suspect A was the real guilty suspect, while Suspects B and C could be ruled out because of the exonerating clues. However, each group member was not privy to all of the relevant information. To uncover the truth, individuals had to exchange and combine their knowledge with the knowledge of their group members, as the common information incorrectly indicated that Suspect B or C was guilty.

### Procedure

To start the procedure, participants were given 3 min to rest (*Figure 8D*). They sat around a table without partitions between each other. Building on prior group decision-making research (*Stasser and Stewart, 1992*; *Xie et al., 2023a*), we refined all stages to enhance controllability throughout the process (i.e., a. Reading information, b. Sharing private information, c. Discussing information, d. Decision). Each group member was then given 18 common and 2 private pieces of information, which they read in 5 min (*Figure 8D*). Each participant would get a piece of paper, which presented the information. Participants could read independently. Subsequently, each triad was required to complete verbal information exchange, comprising both group sharing and group discussion (*Figure 8D*; *Xie et al., 2023b*). During the group sharing, participants entered Tencent Meeting via their mobile phones and were able to text their private information in the chat box to their group members for 5 min. During the group discussion, each group discussed the information that had been disclosed orally for 20 min. Participants were sitting and communicating around a table. The distance between adjacent participants was about 15 cm, and the distance between face-to-face participants was about 40 cm. In this process of discussion, the participants were able to communicate face-to-face and verbally. After discussion, all triads were given 5 min to answer the following questions: (1) the probability of three suspects, 0–100% for each suspect; (2) the motivation and tool of crime; and (3) deduced the entire process of crime. The three questions were presented on paper, allowing participants to write their answers directly on the same sheet. Subsequently, three independent raters used these paper questionnaires to record and calculate the scores for each group.

### fNIRS data acquisition

In this study, the brain activities of participants in each group were simultaneously recorded with fNIRS using an ETG-7100 optical topography system (Hitachi Medical Corporation, Japan). The absorption of near-infrared light (two wavelengths: 695 and 830 nm) was measured with a sampling rate of 10 Hz. The oxyhemoglobin (HbO) and deoxyhemoglobin (HbR) were obtained under the modified Beer–Lambert law. We focused our analyses on the HbO signal for the following reasons: (1) HbO concentration is sensitive to changes in regional cerebral blood flow (*Hoshi, 2003*); (2) the HbO signal was reported to have a higher signal-to-noise ratio than the HbR signal (*Mahmoudzadeh et al., 2013*); and (3) an increasing number of studies have revealed neural synchronization based on the HbO signal (*Yang et al., 2020*).

Two optode probe sets were used to cover each participant's prefrontal and left TPJ regions (*Figure 8—figure supplement 1*). The DLPFC plays a crucial role in group decision-making processes, with findings suggesting that individuals exhibiting reduced prefrontal activity were more prone to out-group exclusion and demonstrated stronger in-group preferences (*Goupil et al., 2021*; *Jankovic, 2014*; *Yang et al., 2020*). Similarly, the left TPJ has been previously reported to be associated with decision-making and information exchange (*De Freitas et al., 2019*; *Tindale and Winget, 2019*). For each participant, one 3 × 5 optode probe set (8 emitters and 7 detectors forming 22 measurement points with 3 cm optode separation, see *Supplementary file 1, table S1* for detailed MNI coordinates) was placed over the prefrontal cortex (reference optode is placed at Fpz, following the international 10–20 system for positioning). The other 2 × 4 probe set (4 emitters and 4 detectors forming 10 measurement points with 3 cm optode separation, see *Supplementary file 2, table S2* for detailed MNI coordinates) was placed over the left TPJ (reference optode is placed at T3, following the international 10–20 system for positioning). The probe sets were examined and adjusted to ensure consistency of the positions across the participants. After the completion of data collection, we utilized the Vpen positioning system to accurately locate the detection light poles, ultimately obtaining the MNI positioning coordinates.

## Behavioral analyses

### Group identification

To examine and quantify the manipulation of group identification, the scores of the participants in each triad were averaged to determine the group identification, and an ANOVA with repeated measures was performed to examine group identification, with the levels of group identification (High/Low) serving as a between-subjects variable, and the orders of rating group identification (group identification_1/group identification_2) as a within-subjects variable (*Figure 8C*). Moreover, Pearson's correlation was used to examine the relationship between group identification_2 and collective performance.

### Individual performance

Individual performance was evaluated based on participants' accuracy in solving the case, including their probability estimates for the three suspects (0–100% each, 2 points per suspect, totaling 6 points), identification of the crime's motivation (1 point) and tool (1 point), and deduction of the full crime process (20 points). Three independent raters assessed each participant's performance, demonstrating high consistency (Cronbach's $\alpha$ = 0.89).

### Collective performance

Collective performance was calculated by averaging the individual scores assigned by the three raters for each group, representing the group's overall accuracy in the case solution.

An independent *t*-test was conducted to examine collective performance, with group identification levels being the independent variable and collective performance as the dependent variable. Moreover, we employed a regression model to examine how varying levels of group identification affect collective performance, using group identification scores as the independent variable and collective performance as the dependent variable.

### The similarity in individual-collective performance

After the *z*-score normalization of each item for individual and collective performance, we calculated the Euclidean distance (*Equation 1*) between individual and collective performance. In *Equation 1*, $x$ is the individual score, $y$ is the collective score ($y$ is calculated from the three per capita scores), and $i$ stands for the group number. So, $x_i$ means the individual score of participants in the $i$ group, and $y_i$ means the collective score of the $i$ group. $d(x, y)$ represents the distance from the individual to the collective score. A smaller distance indicated a higher similarity in individual-collective performance, while a larger distance suggested a lower similarity in individual-collective performance,

$$d\left(x, y\right) = \sqrt{\sum \left(x_i - y_i\right)}. \tag{1}$$

To investigate whether there was a significant difference in the similarity in individual-collective performance between conditions, an independent *t*-test was conducted with group identification levels as the independent variable and the similarity in individual-collective performances as the dependent variable.

## fNIRS data analyses

### Overview

We aimed to investigate the neural mechanisms underlying the impact of different levels of group identification on collective performance. (1) To do this, we sought to examine whether the individual differences in individual performance were reflected in single-brain activations. We first identified task-related brain regions and compared the single-brain activation of different levels of group identification. We then examined the correlation between single-brain activation and individual performance and tested whether the relationship between an individual's perceived group identification and individual performance was mediated by single-brain activation. (2) We sought to examine whether the group differences in collective performance were reflected in within-GNS. We first identified task-related GNS and compared GNS of different levels of group identification. We then examined the correlation between collective performance and GNS and tested whether the relationship between the group identification score of each triad and collective performance was mediated by GNS. (3)

Examining brain activation connectivity, we sought to bridge single-brain activations and the corresponding GNS, thus unifying individual decision-making and collective performance. We first identified task-related brain activation connectivity and compared brain activation connectivity of different levels of group identification. We then examined the correlation between the similarity in individual-collective performance and brain activation connectivity and tested whether the relationship between the individual's single-brain activations and the corresponding GNS was mediated by brain activation connectivity.

## Pre-processing approach

We sought to explore the neural mechanisms that manipulated group identification and its effect on collective performance. Data were preprocessed using the Homer2 package in MATLAB 2020b (Mathworks Inc, Natick, MA, USA). First, motion artifacts were detected and corrected using a discrete wavelet transformation filter procedure. After that, the raw intensity data were converted to optical density (OD) changes. Then, kurtosis-based wavelet filtering (Wav Kurt) was applied to remove motion artifacts with a kurtosis threshold of 3.3 (*Chiarelli et al., 2015*). Based on a prior multi-brain study of social interactions (*Cheng et al., 2022*), the output was bandpass filtered using a Butterworth filter with order 5 and cut-offs at 0.01 and 0.5 Hz to remove longitudinal signal drift and instrument noise. Finally, OD data were converted to HbO concentrations.

## Single-brain activation

The detailed steps are as follows: (1) We analyzed the data using SPM-based software (*Ye et al., 2009*), focusing on the time series from information reading to decision-making, and extracted the HbO signal for each participant and triad. The onsets and durations of these time series were used to generate the stimulus design, which was convolved with a canonical hemodynamic response function (HRF) using NIRS-SPM. (2) We then employed a general linear model (GLM) to fit the predicted signals to the actual data, obtaining beta estimates (regression coefficients) for each parameter in the single-subject design matrices. Within the GLM framework, contrast maps were generated to compare activation between the baseline (resting phase) and task sessions (encompassing information reading to decision-making) for each subject, based on the HbO signal. Next, the regression factor was convolved with the HRF, and regression analysis provided brain activation values representing decision-making activity for each participant across all task stages and channels. For group-level analysis, we conducted second-level, random-effects analyses (*Friston et al., 2007*) by aggregating the $\beta$ values across participants. A one-sample *t*-test was performed for each channel to determine whether mean activation significantly exceeded zero (i.e., greater task-related activation than rest), with p-values corrected for FDR (p < 0.05, *Benjamini and Hochberg, 1995*). (3) Channels showing significant single-brain activations were identified as task-related and included in further analyses, and the results were visualized on a standard MNI brain template using summary statistics.

Subsequently, we conducted independent *t*-tests on single-brain activations in task-related brain regions, with group identification levels as the independent variable. The p-values were also adjusted to control for FDR (p < 0.05; *Benjamini and Hochberg, 1995*). Then, we used Pearson's correlation analyses to investigate the relationship between single-brain activation and individual performance (i.e., evaluated based on participants' accuracy in solving the case). Finally, we employed the PROCESS model 4 to construct a mediation model with 5000 bootstrap resamples (*Preacher and Hayes, 2008*) to examine the relationship between an individual's perceived group identification and individual performance, which was mediated by single-brain activation.

## Within-GNS

After pre-processing, GNS was used as the neural index (i.e., interpersonal brain activities that co-vary along the time course). Concerning GNS, and similar to previous studies (*Pan et al., 2021*; *Yang et al., 2020*), the wavelet transform coherence (WTC) (*Equation 2*) was used to assess the cross-correlation between two oxy-Hb time series of pairs of participants. Here, *t* denotes the time, *s* indicates the wavelet scale, ⟨·⟩ represents a smoothing operation in time, and *W* is the continuous wavelet transform (*Grinsted et al., 2004*). Within each triad (taking one triad with subject IDs of 1, 2, and 3 as an example), WTC was applied to generate the brain-to-brain coupling of each pair in each triad (Coherence 1&2, Coherence 1&3, and Coherence 2&3). Then, three coherence values from three

pairs were averaged as the GNS for each triad, that is, GNS = (Coherence 1&2 + Coherence 1&3 + Coherence 2&3)/3,

$$WTC\left(t,s\right) = \frac{\left|\left\langle s^{-1}w^{ij}\left(t,s\right)\right\rangle\right|^2}{\left|\left\langle s^{-1}w^{i}\left(t,s\right)\right\rangle\right|^2\left|\left\langle s^{-1}w^{j}\left(t,s\right)\right\rangle\right|^{2'}}.$$ (2)

Regarding the first step, we estimated whether GNS was enhanced during the task compared to the baseline. Time-averaged GNS (also averaged across channels in each group) was compared between the baseline session (i.e., the resting phase) and the task session (from reading information to making decisions) using a series of one-sample *t*-tests. Here, p-values were thresholded by controlling for FDR (p < 0.05; *Benjamini and Hochberg, 1995*). When determining the frequency band of interest, the time-averaged GNS was also averaged across channels. After that, we analyzed the time-averaged GNS of each channel. Then, channels showing significant GNS were regarded as regions of interest and included in subsequent analyses. An independent *t*-test was conducted on GNS, with group identification levels being the independent variable. Here, p-values were thresholded by controlling for FDR (p < 0.05; *Benjamini and Hochberg, 1995*). After that, the nonparametric permutation test was conducted on the observed interaction effects on GNS of the real group against the 1000 permutation samples. By pseudo-randomizing the data of all participants, a null distribution of 1000 pseudo-groups was generated (e.g., time series from member 1 in group 1 were grouped with member 2 in group 2 and member 3 in group 3). The GNS of 1000 reshuffled pseudo-groups was computed, and the GNS of the real groups was assessed by comparing it with the values generated by 1000 reshuffled pseudo-groups. To provide a complete picture of the underlying neural features, we also analyzed the GNS based on the HbR signal (see Supplementary Materials). Second, the Pearson's correlation between GNS and collective performance (i.e., calculated by averaging the individual scores assigned by the three raters for each group) was performed. It is plausible that neural synchronization is closely associated with group identification and collective performance, suggesting that it serves as a promising mechanism to explore how group identification influences collective outcomes. Moreover, previous research has established that neural synchronization facilitates the emergence of group identification, and the degree to which neural synchronization occurs among group members may shape how individuals identify with the group (*Xie et al., 2023a*; *Reinero et al., 2021*). Ultimately, PROCESS model 4 with 5000 bootstraps resamples was used to test how GNS mediated the relationship between group identification and collective performance (*Preacher and Hayes, 2008*).

## The brain activation connectivity

Studies in neuroimaging have indicated that brain activation connectivity could be useful in understanding brain functional integration (*Lu et al., 2010*; *Montero-Hernandez et al., 2018*; *Yang et al., 2020*). Consequently, an exploratory analysis was conducted to test the hypothesis that brain activation connectivity could support the connection between an individual's single-brain activations and the corresponding group's GNS, thereby linking individual decision-making and collective performance.

To explore this hypothesis, we first isolated HbO brain activity associated with individual and collective performance. Following that, we analyzed Pearson's correlations between the original HbO data in the region related to individual and collective performance, denoted as brain activation connectivity (*Lu et al., 2010*). Subsequently, we carried out one-sample *t*-tests on brain activation connectivity to ascertain if there was any connection to the task. Furthermore, independent *t*-tests were conducted on brain activation connectivity with the group identification levels as the independent variable, accounting for the FDR (p < 0.05; *Benjamini and Hochberg, 1995*). Finally, we employed correlation and mediation analyses to assess if brain activation connectivity could explain the connection between individuals' single-brain activation and the related group's GNS. We examined the connection between the similarity in individual-collective performance and the correlation of brain activation, as well as whether the impact of each individual's single-brain activation on the corresponding group's GNS was regulated by their brain activation connectivity. We utilized the PROCESS tool in SPSS to investigate the proposed moderation effect. Specifically, we applied Model 1 with 5000 bootstrap resamples to examine the interaction between the independent variable (i.e., single-brain activation) and the moderator (i.e., brain activation connectivity) in predicting the dependent variable (i.e., GNS).

It is noteworthy that prior to analysis, all variables in the moderation model were mean-centered to reduce multicollinearity and improve the interpretability of interaction terms.

### Dynamic analyses

Our goal was to gain a more thorough understanding of how group identification influences collective performance through neural processes, with a focus on dynamic perspectives and tracking. We used 1-min epochs to analyze the average single-brain activation, GNS, and brain activation connectivity during the task. We then plotted the time course of dynamic single-brain activation and conducted one-sample *t*-tests to identify significant single-brain activation, GNS, and the correlation of brain activation periods. To better understand the interactive behavior revealed by dynamic single-brain activation, we linked the brain activation time series with video recordings of interactive behavior.

### Additional modal measures and analyses

Additionally, we aimed to gain a more comprehensive understanding of how various group identifications influence collective performance. To this end, we obtained more evidence through other methods. Previous studies have shown that the quality of information exchange, such as verbal interactions, eye contact, and smiling, is a reliable indicator of group behavior and is associated with collective performance (*Dikker et al., 2022*; *Hirsch et al., 2018*; *Jiang et al., 2015*; *Liu et al., 2021*; *Xie et al., 2023b*). Therefore, three independent raters were asked to rate the quality of the information exchange of each group, with a Cronbach's alpha of 0.85. The raters were guided to consider verbal interactive frequency (e.g., 'I agree with you', 'You're right', 'I understand what you mean') and nonverbal interactive frequency (e.g., eye contact and smiling) (*Jiang et al., 2015*; *Xie et al., 2023a*; *Figure 7A*). The evaluation period consisted of group sharing and discussion, in which the raters evaluated the suggested items in 1-min increments. The scores of the 25 periods were then compiled. The final quality of the group's information exchange was determined by taking the average of the scores of the three raters for each group. A higher quality of the group's information exchange entails communicating more fully.

Initially, independent *t*-tests were performed to examine the quality of information exchange between the group identification levels, which was the independent variable. Subsequently, Pearson's correlation between the quality of information exchange and collective performance was assessed.

Behavioral metrics (i.e., the quality of information exchange) offer direct indicators of the strategies and patterns individuals use during the decision-making process. Neural data (i.e., GNS) can uncover the neural activity linked to decision-making processes in the brain. For a more comprehensive insight into the processes involved in group decision-making, we performed the hierarchical multiple regression analysis with collective performance as the dependent variable to determine the weight of prediction of the quality of information exchange, GNS, and their interaction.

## Acknowledgements

This work is supported by the National Natural Science Foundation of China (71942001), STI 2030 – Major Projects (2021ZD0200500), the National Natural Science Foundation of China (32071082), Key Specialist Projects of Shanghai Municipal Commission of Health and Family Planning (ZK2015B01), and the Programs Foundation of Shanghai Municipal Commission of Health and Family Planning (201540114).

## Additional information

### Funding

| Funder | Grant reference number | Author |
| --- | --- | --- |
| The National Natural Science Foundation of China | 71942001 | Xianchun Li |
| STI 2030-Major Projects | 2021ZD0200500 | Xianchun Li |

| Funder | Grant reference number | Author |
|---|---|---|
| The National Natural Science Foundation of China | 32071082 | Xianchun Li |
| Key Specialist Projects of Shanghai Municipal Commission of Health and Family Planning | ZK2015B01 | Xianchun Li |
| Programs Foundation of Shanghai Municipal Commission of Health and Family Planning | 201540114 | Xianchun Li |

The funders had no role in study design, data collection, and interpretation, or the decision to submit the work for publication.

## Author contributions

Enhui Xie, Conceptualization, Data curation, Formal analysis, Methodology, Writing - original draft; Shuyi Zha, Data curation, Methodology, Writing - review and editing; Yiyang Xu, Data curation, Writing - review and editing; Xianchun Li, Conceptualization, Funding acquisition, Writing - review and editing

## Author ORCIDs

Enhui Xie ⓘ https://orcid.org/0000-0002-4939-9791
Xianchun Li ⓘ https://orcid.org/0000-0002-0744-5462

## Ethics

The study had full ethical approval by the University Committee on Human Research Protection (HR2-0189-2022), East China Normal University. Informed written consent was obtained from each participant before each experiment.

Reviewer #1 (Public review): https://doi.org/10.7554/eLife.100000.4.sa1
Author response https://doi.org/10.7554/eLife.100000.4.sa2

# Additional files

## Supplementary files

Supplementary file 1. MNI coordinate Position of 3×5 optode probe set.

Supplementary file 2. MNI coordinate Position of 2×4 optode probe set.

Supplementary file 3. The results of HbR.

## Data availability

Data of this project were obtained from the GitHub repository (available at https://github.com/xehui/group-identification, *xehui, 2024*).

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
