## [Editor Report · eLife Assessment]

This timely and **important** study used functional near-infrared spectroscopy hyperscanning to examine the neural correlates of how group identification influences collective behavior. The work provides **solid** evidence to indicate that the synchronization of brain activity between different people underlies collective performance and that changes in brain activity patterns within individuals may, in turn, underlie this between-person synchrony, although the order in which different task stages were completed could not be counter-balanced. This study will be of interest to researchers investigating the neuroscience of social behavior.

---

## [Referee Report · Reviewer #1 (Public review)]

The article provides a timely and well-written examination of how group identification influences collective behaviors and performance using fNIRs and behavioral data.

Strengths:

(1) Timeliness and Relevance:

The topic is highly relevant, particularly in today's interconnected and team-oriented work environments. Triadic hyperscanning is important to understand group dynamics, but most previous work has been limited to dyadic work.

(2) Comprehensive Analysis:

The authors have conducted extensive analyses, offering valuable insights into how group identification affects collective behaviors.

(3) Clear Writing:

The manuscript is well-written and easy to follow, making complex concepts accessible.

Comments on previous revisions:

Most reviewer concerns have been addressed in the revised manuscript, but some limitations persist with respect to core aspects of study design, such as the long block durations and lack of counter-balancing.

---

## [Author Response]

The following is the authors’ response to the previous reviews

We are appreciative of the reviewers’ and editors’ constructive suggestions of manuscript, which have helped us to improve our manuscript. We have made considerable revisions to our details of data analyses.

The reason that the reviews did not change is that there were really three central points that led to the "incomplete". These were (1) the fact that there was potentially a selection bias due to double dipping, and (2) there was potentially a time-confound due to the lack of counterbalancing (3) There is confusion about how the modeling was done, but it seems like the modelling was of the complete block (rather than tied to specific events in that block).(1) Double dipping

We appreciate the opportunity to explain our robust safeguards against double-dipping and have provided detailed clarifications regarding the data analyses (pp.11-14).Our study ensures statistical independence between task-related region selection and hypothesis testing through three orthogonal mechanisms:

(1) Regressor Orthogonality:Statistical Independence Between Selection and Testing

The selection regressor (group mean activation) was mathematically independent from test regressors (group differences, behavioral scores). This was confirmed through our GLM implementation: First-level: Task vs. rest contrast (*β* values) for each participant; Second-level: One-sample *t*-tests (selection) vs. independent group/behavioral tests.

(2) Multimodal Validation: Complementary Neural and Behavioral Measures

We employed multiple distinct metrics to provide convergent yet independent validation of effects.

Neural Measures: Three orthogonal indices assessed different neural dimensions.

A. Single-brain activation examines neural activity patterns within individual decision-makers,

B. while within-group neural synchronization (GNS) quantifies the temporal alignment of neural activity across interacting group members during shared decision processes.

C. Functional connectivity (FC) analyses, by contrast, measure correlated activity between different brain regions within individual participants.

Behavioral Safeguards: Behavioral metrics were analyzed in independent regressions, avoiding circularity.

A. Individual performance was based on personal accuracy,

B. collective performance represented the group-level average accuracy across raters, and

C. their similarity was quantified as the Euclidean distance between individual and collective scores.

(3) Statistical Safeguards

We further ensured independence by applying strict FDR correction at both selection (*p* < 0.05) and testing stages (*p* < 0.05). Besides, permutation test was conducted, we tested 1,000 pseudo-group iterations for GNS null distributions.

Drawing on both classic and latest NIRS (e.g., Jiang et al., 2015; Liu et al., 2023; Stolk et al., 2016; Xie et al., 2023) and NIRS hyperscanning studies (e.g., Liu et al., 2019; P’arnamets et al., 2020; Reinero et al., 2021; Számadó et al., 2021; Solansky, 2011), we performed the data analyses. Below, we provide the details of our data analysis:

Single-brain activation. To identify task-related brain regions (channels), we used a one-sample *t*-test based on brain activation data from all participants during the task compared to the baseline (resting state).

(1) Data Collection: Each participant had brain activation data (HbO signals measured by fNIRS) during the task (the entire process of reading, sharing, discussing, and decision-making) and the resting state (baseline).

(2) Pre-processing: We sought to explore the neural mechanisms that manipulated group identification and its effect on collective performance. Data were preprocessed using the Homer2 package in MATLAB 2020b (Mathworks Inc, Natick, MA, USA). First, motion artifacts were detected and corrected using a discrete wavelet transformation filter procedure. After that, the raw intensity data were converted to optical density (OD) changes. Then, kurtosis-based wavelet filtering (Wav Kurt) was applied to remove motion artifacts with a kurtosis threshold of 3.3 (Chiarelli, Maclin, Fabiani, & Gratton, 2015). Based on a prior multi-brain study of social interactions (Cheng et al., 2022), the output was bandpass filtered using a Butterworth filter with order 5 and cut-offs at 0.01 and 0.5 Hz to remove longitudinal signal drift and instrument noise. Finally, OD data were converted to HbO concentrations.

(3) Individual-Level Analysis: First, a GLM was used to compute the "task vs. rest" brain activation contrast for each participant [0,1], obtaining each individual's "task effect" value (*β* value, representing task activation strength).

(4) Group-Level Analysis: These "task effect" values from all participants were then aggregated, and a one-sample *t*-test was performed for each brain region (or channel) to determine whether the average activation in that region was significantly greater than 0 (i.e., significantly more active during the task compared to the resting state).

(5) Task-Related Regions: If the *t*-test result for a brain region was significant (p < 0.05, FDR-corrected), we considered that region "task-related" and suitable for further analysis.

(6) Subsequent Tests:

- Group Comparisons: We examined differences in activation between groups (e.g., high vs. low group identification) using independent t-tests on the same task vs. baseline contrast.

- Behavioral Correlations: We analyzed relationships between task-related activation (*β* values) and behavioral scores (e.g., individual performance) using Pearson analyses.

- Mediation model: We examined the relationship between an individual's perceived group identification and individual performance, which was mediated by task-related activation (*β* values).

Within-Group Neural Synchronization (GNS).

(1) Data Collection and Pre-processing as above

(2) Calculation: WTC was applied to generate the brain-to-brain coupling of each pair in each triad (Coherence1&2, Coherence 1&3, and Coherence 2&3). Then, three coherence values from three pairs were averaged as the GNS for each triad, that is, GNS = (Coherence 1&2 + Coherence 1&3 + Coherence 2&3) / 3.

(3) Task-Related Regions: Time-averaged GNS (also averaged across channels in each group) was compared between the baseline session (i.e., the resting phase) and the task session (from reading information to making decisions) using a series of one-sample t-tests. When determining the frequency band of interest, the time-averaged GNS was also averaged across channels. After that, we analyzed the time-averaged GNS of each channel. Then, channels showing significant GNS were regarded as regions of interest and included in subsequent analyses.

(4) Permutation test: The nonparametric permutation test was conducted on the observed interaction effects on GNS of the real group against the 1,000 permutation samples.

(5) Subsequent Tests:

- Group Comparisons: We examined differences in activation between groups (e.g., high vs. low group identification) using independent *t*-tests on the same task vs. baseline contrast.

- Behavioral Correlations: The Pearson’s correlation between GNS and collective performance (i.e., calculated by averaging the individual scores assigned by the three raters for each group) was performed.

- Mediation model: We examined how GNS mediated the relationship between group identification and collective performance.

The brain activation connectivity.

(1) Data Collection and Pre-processing as above

(2) Calculation: Exploratory Pearson’s correlations between individual performance related HbO and collective performance-related HbO.

(3) Moderation analysis: Single-brain activation × connectivity → GNS.

(2) Counterbalancing.

We sincerely appreciate this valuable methodological insight. Building on prior group decision-making research (De Wilde et al., 2017; Stasser et al., 1992), we refined all stages to enhance experimental control and procedural clarity throughout the process (i.e., a. Reading information, b. Sharing private information, c. Discussing information, d. Decision) (Xie et al., 2023). Importantly, we maintained a fixed task sequence to preserve ecological validity, as this progression mirrors natural group decision-making dynamics.

While this design choice precludes sequential counterbalancing, several factors mitigate potential temporal confounds: (1) random assignment and uniform task timing across conditions minimize systematic between-group differences; (2) our whole-block GLM approach captures sustained decision-related neural activity rather than phase-specific effects; and (3) We fully acknowledge this limitation and will incorporate a detailed discussion of temporal considerations in the revised manuscript, while noting that our design provides unique advantages for studying naturalistic decision-making processes.

(3) The modelling was of the complete block

In our revised manuscript, we have explicitly stated that the analysis was performed at the block level rather than the event level, for the following reasons:

(1) The hidden profile task is inherently a “group decision-making process” that unfolds dynamically across multiple stages (reading, sharing, discussing, and deciding). Prior research in this paradigm (De Wilde et al., 2017; Stasser & Titus, 1985; Xie et al., 2023) has consistently treated these phases as integrated blocks because the key cognitive and social processes (e.g., information integration, deliberation, and consensus formation) occur over extended interactions rather than discrete events.

(2) Methodologically, our fNIRS hyperscanning approach requires longer blocks to reliably capture the slow hemodynamic response and the gradual emergence of inter-brain neural synchronization during naturalistic social exchanges (Cui et al., 2012; Liu et al., 2019). Event-related designs, while useful for transient stimuli, are less suited for studying prolonged, interactive decision-making where neural coupling develops over time. Thus, our block-based analysis aligns with both the cognitive demands of the task and the neuroimaging constraints, ensuring robust detection of group-level neural dynamics.